# Anionic Pulmonary Surfactant Lipid Treatment Inhibits Rhinovirus A Infection of the Human Airway Epithelium

**DOI:** 10.3390/v15030747

**Published:** 2023-03-14

**Authors:** Mari Numata, Satria Sajuthi, Yury A. Bochkov, Jessica Loeffler, Jamie Everman, Eszter K. Vladar, Riley A. Cooney, Richard Lee Reinhardt, Andrew H. Liu, Max A. Seibold, Dennis R. Voelker

**Affiliations:** 1Department of Medicine, National Jewish Health, Denver, CO 80206, USA; 2Center for Genes, Environment and Health, National Jewish Health, Denver, CO 80206, USA; 3Department of Pediatrics, University of Wisconsin School of Medicine and Public Health, Madison, WI 53792, USA; 4Department of Medicine, University of Colorado School of Medicine, Aurora, CO 80045, USA; 5Department of Immunology and Genomic Medicine, National Jewish Health, Denver, CO 80206, USA; 6Section of Pediatric Pulmonary & Sleep Medicine, Children’s Hospital Colorado and University of Colorado School of Medicine, Aurora, CO 80045, USA; 7Department of Pediatrics, National Jewish Health, Denver, CO 80206, USA

**Keywords:** antiviral, innate immunity, primary human airway epithelial cells, replication organelle, human phosphatidylinositol 4-kinases

## Abstract

Rhinoviruses (RVs) are major instigators of acute exacerbations of asthma, COPD, and other respiratory diseases. RVs are categorized into three species (RV-A, RV-B, and RV-C), which comprise more than 160 serotypes, making it difficult to develop an effective vaccine. Currently, no effective treatment for RV infection is available. Pulmonary surfactant is an extracellular complex of lipids and proteins that plays a central role in regulating innate immunity in the lung. The minor pulmonary surfactant lipids, palmitoyl-oleoyl-phosphatidylglycerol (POPG) and phosphatidylinositol (PI), are potent regulators of inflammatory processes and exert antiviral activity against respiratory syncytial virus (RSV) and influenza A viruses (IAV). In the current study, we examined the potencies of POPG and PI against rhinovirus A16 (RV-A16) in primary human airway epithelial cells (AECs) differentiated at an air–liquid interface (ALI). After AECs were infected with RV-A16, PI reduced the viral RNA copy number by 70% and downregulated (55–75%) the expression of antiviral (MDA5, IRF7, and IFN-lambda) and CXCL11 chemokine genes. In contrast, POPG only slightly decreased MDA5 (24%) and IRF7 (11%) gene expression but did not inhibit IFN-lambda gene expression or RV-A16 replication in AECs. However, both POPG and PI inhibited (50–80%) IL6 gene expression and protein secretion and CXCL11 protein secretion. PI treatment dramatically attenuated global gene expression changes induced by RV-A16 infection alone in AECs. The observed inhibitory effects were indirect and resulted mainly from the inhibition of virus replication. Cell-type enrichment analysis of viral-regulated genes opposed by PI treatment revealed the PI-inhibited viral induction of goblet cell metaplasia and the virus-induced downregulation of ciliated, club, and ionocyte cell types. Notably, the PI treatment also altered the ability of RV-A16 to regulate the expression of some phosphatidylinositol 4-kinase (*PI4K*); acyl-CoA-binding, domain-containing (*ACBD*); and low-density lipoprotein receptor (*LDLR*) genes that play critical roles in the formation and functioning of replication organelles (ROs) required for RV replication in host cells. These data suggest PI can be used as a potent, non-toxic, antiviral agent for RV infection prophylaxis and treatment.

## 1. Introduction

Viral infections cause approximately 80% of asthma exacerbations in children and adults [1]. Rhinoviruses (RVs) are the most frequent causes of virus-induced exacerbations of asthma [2,3,4,5]. RVs are categorized into three species (RV-A, RV-B, and RV-C) that include more than 160 types [6,7,8]. RV infections are significantly associated with hospitalization for acute asthma exacerbations [1,9,10]. RV-A and RV-C can induce more severe illnesses [4,11] and have been strongly correlated with a high risk of developing asthma in children [10,12] compared to RV-B. The combination of allergic sensitization to respiratory allergens and viral infections is determined as a high-risk factor for asthma development and asthma exacerbations [10,11,13]. These studies suggest that prophylaxis and/or treatments for RV will be of high clinical importance in preventing childhood asthma and its outcomes. Despite decades of research and clinical trials, there are currently no approved vaccines or antivirals for the prevention or treatment of RV [2,10].

The pulmonary surfactant system of the lung is a mixture of lipid and protein complexes that regulates both the biophysical properties of the alveolar compartment and the innate immunity of the lung. The major components of the pulmonary surfactant system are phospholipids, and dipalmitoyl-PC (DPPC) belongs to a major class of phosphatidylcholine (PC) that is responsible for reducing surface tension and preventing lung collapse [14]. There are two minor phospholipids, phosphatidylglycerol (PG) and phosphatidylinositol (PI), that exist in the alveolar compartment at a very high concentration compared to other organs [14]. Palmitoyl-oleoyl-phosphatidylglycerol (POPG) is the major molecular species of PG, and dioleoyl-phosphatidylinositol is the major molecular species of PI. The roles of these lipids have been enigmatic, but recent studies have shown that POPG and PI play critical roles in regulating the innate immunity of the lung [14,15,16]. Our findings have demonstrated that POPG and PI antagonize multiple toll-like receptors (TLRs) (TLRs 1, 2, 3, 4, and 6) and inhibit the inflammatory responses and proinflammatory cytokine production induced by TLR activation [14,15,17]. 

We have kept accumulating evidence that both POPG and PI have very potent anti-viral effects against several respiratory RNA viruses, including respiratory syncytial virus (RSV) and influenza A viruses (e.g., pH1N1-IAV (H1N1 A/California/07/2009, H1N1-PR8 (H1N1 influenza A/PR/8/34), and H3N2-IAV (H3N2/Philippines 82)), in vitro and in several in vivo animal models [18,19,20,21]. These findings led us to examine whether POPG or PI can antagonize human RV infection. In this report, we aimed to determine whether POPG and PI can attenuate inflammatory responses induced by rhinovirus A16 (RV-A16) and inhibit its replication in differentiated primary human airway epithelial cells (AECs), and to study the possible mechanisms of POPG and PI antagonism against RV-A16 replication. 

## 2. Material and Methods

### 2.1. Cell Culture and Virus

Primary AECs were isolated by the National Jewish Health (NJH) Live Cell Core from human lung specimens that were obtained from de-identified donors whose lungs were not suitable for transplantation by the International Institute for the Advancement of Medicine (Edison, NJ, USA) and Donor Alliance of Colorado. The NJH Institutional Review Board (IRB) approved the research on lung cells (HS-3209) and the use and collection of cells and tissues by NJH Live Cell Core (HS-2240). Cells were differentiated and maintained using air–liquid interface (ALI) culture. We routinely performed three individual experiments using primary tracheal epithelial cells obtained from four different non-smoking donors. Tracheal epithelial cells were expanded on collagen-coated plates (Corning), initially using tracheal epithelial cell growth media (BEGM bullet kit from LONZA, (Walkersville, MD, USA)), and subsequently cultured at ALI in 24-well plates with transwell inserts (Corning, Kennebunk, ME, USA), as previously reported [22]. After the shift to ALI conditions, the cultures underwent differentiation for 14–16 days. H1-HeLa cells were obtained from Dr. Wai-Ming Lee (University of Wisconsin School of Medicine and Public Health, Madison, WI, USA) and cultured in DMEM/F12 medium (GIBCO, New York, NY, USA) supplemented with 1% glutamine, 15 mM of HEPES, 1% penicillin and streptomycin (GIBCO, New York, NY, USA), and 10% heat-inactivated bovine growth serum ((BGS), Cytiva HyClone, S Logan, UT, USA). RV-A16 was obtained from the ATCC (ATCC# VR-283), propagated in H1-HeLa cells, and purified through ultracentrifugation [23,24]. Viral titers were quantified using plaque assays, as described [23,24].

### 2.2. Pulmonary Surfactant Lipid Treatment and Viral Infection of Differentiated AEC Cultures

Phospholipids (Avanti, Birmingham, AL, USA) were prepared as unilamellar liposomes through bath sonication, as previously described [18,19,20]. To examine the anti-RV effects of phospholipids (POPG, PI) with inhibitory effects against other RNA viruses or the control lipid palmitoyl-oleoyl-phosphatidylcholine (POPC), liposomes were added to cells apically (25 µL/well) and incubated for 16 h at the concentration of 10 mg/mL (POPG), 4 mg/mL (PI), and 20 mg/mL (POPC). Subsequently, RV-A16 was added apically at 2 × 10^4^ plaque-forming units (pfu) per/well for 4 h at 35 °C with or without lipids in a total volume of 50 μL. Unbound virions were removed by washing with Dulbecco’s phosphate-buffered saline (DPBS) three times, and the epithelial cells were next incubated for 48 h at 35 °C either with or without phospholipids in 25 μL of the apical medium. At 48 h, the apical medium was harvested to quantify the secretion of IL-6 (e-Bioscience, San Diego, CA, USA) and CXCL-11 (Peprotech, Rocky Hill, NJ, USA) by ELISA.

### 2.3. Quantification of RV-A16 Genomic Copies and Host Gene Expression using qRT-PCR

RNA extractions from cultured cells were performed with the RNeasy Mini kit (Qiagen, Hilden, Germany). The viral RNA levels were determined using qRT-PCR and FAST SYBR Green master PCR mix (Thermo Fisher Scientific, Waltham, MA, USA), as previously described [25], and normalized to human beta-actin RNA levels [26]. We also determined the expression of melanoma differentiation-associated protein 5 (*MDA5*), interferon regulatory factor 7 (*IRF7*), and interferon lambda (IFN-λ) genes using qRT-PCR, as described in [26,27]. The TaqMan gene expression assay for human CXCL11 was obtained from Thermo Fisher Scientific [28]. The housekeeping gene, beta-glucuronidase (GUSB), was used as an internal control to normalize the expression of the CXCL11 (*CXCL11)* gene. The primer sequences for each gene are shown in Appendix A.

### 2.4. Whole-Transcriptome RNA Sequencing and Differential Expression Analysis

Library preparation and RNA sequencing were conducted at National Jewish Health (Seibold Laboratory, Denver, CO, USA). Briefly, RNA-seq libraries were generated on replicate ALI samples obtained from three tracheal donors and treated in the following ways: (1) mock (shows as uninfected (UN)), (2) lipid only (PI, POPG, or PC), (3) RV-A16 alone at 4 h and 48 h (RV16 4 h, RV16 respectively), or (4) lipid pretreatment followed by RV-A16 infection (RV16 + PI, RV16 + POPG, RV16 + PC). KAPA mRNA Hyper Prep (Roche, Indianapolis, IN, USA) whole-transcriptome libraries were constructed on the Beckman Coulter FX^P^ automation system with 500 ng of RNA input per sample, and Illumina Dual Index Adapters (IDT Technologies, Coralville, IA, USA) were used to barcode samples using 10 cycles of amplification [29,30]. Raw sequencing reads were trimmed using a skewer (v0.2.2) [31] with the following parameters: end-quality = 15, mean-quality = 25, min = 30. Read alignment to the human reference genome, GRCh38, was performed using HISAT2 (v2.1.0) with default parameters [32]. Gene quantification was performed using htseq-count (v0.9.1) and the GRCh38 Ensemble v84 gene transcript model with the following parameters: stranded = reverse, a = 20, mode = intersection-nonempty [33]. Count data were variance stabilized using the *varianceStabilizingTransformation* function implemented in DESeq2 (v1.22.2). Differential expression analysis was performed using lmerSeq (v0.1.7) [34]. The RNA-seq data were submitted to GEO and assigned under accession number GSE226071.

### 2.5. Pathway and Cell-Type Enrichment Analyses

We used the QIAGEN Ingenuity Pathway Analysis software (IPA v01-21-03) to perform canonical pathway analyses on sets of differentially expressed genes (DEGs) and to produce a cellular graphic of DEGs underlying the *interferon-signaling* pathway enrichment. To identify cell-type enrichments among the genes affected by a viral infection, the effects of which were opposed by PI treatment, we tested for overrepresentation of these gene sets among airway cell-type gene signatures derived from the Human Lung Cell Atlas (HLCA) single-cell project data using hypergeometric tests [35] (https://cellxgene.cziscience.com/collections/6f6d381a-7701-4781-935c-db10d30de293, accessed on 9 December 2022).

### 2.6. Immunofluorescent Staining

For wholemount immunofluorescence, AECs were fixed in 4% paraformaldehyde for 10 min, as previously described [36]. Transwell membranes were cut out of the plastic supports and placed in a humid chamber for staining. Samples were blocked in 10% normal horse serum and 0.1% Triton X-100 in PBS and incubated with primary antibodies against the VP2 protein of RV-A16 and dsRNA overnight at 4 °C, and then with Alexa dye conjugated secondary antibodies and Phalloidin (Thermo Fisher Scientific, Waltham, MA, USA) for 30 min at room temperature. Filters were mounted in Mowiol mounting medium containing 2% N-propyl gallate (Sigma, St. Louis, MO, USA). Samples were imaged with a Zeiss LSM900 confocal microscope. Multiciliated cells, as a fraction of total luminal cells, were determined by quantitating the number of ac. alpha-tubulin-positive cells as a fraction of the total number of luminal cells based on the phalloidin labeling of the apical cell junctions. The antibodies used were: dsRNA (cat. no. MABE1134, Millipore-Sigma, Carlsbad, CA, USA), RV-A16 (cat. no. 18758, QED Bioscience, San Diego, CA, USA), and ICAM1 (cat. no. 14-0549-82, Thermo Fisher, Waltham, MA, USA) [36].

### 2.7. Evaluation of Virus Internalization in AECs using the Trypsinization Method

The differentiated AEC-ALI cells grown in 24-well plates were pre-treated, or not, for 16 h with PI (4 mg/mL) and infected apically with RV-A16 at three various doses (1, 2.5, 5 (×10^5^ pfu/well)) at 4 °C (to enable virus binding to the ICAM1 receptor but to block virus entry to cells) or 35 °C (to enable virus binding and entry) for 2 h with gentle swirling. The cells were washed with DPBS 3 times before lysis with RLT buffer (Qiagen, Germantown, MD, USA) for RNA extraction or treated with 300 µL of 0.25% Trypsin/EDTA (Gibco, Waltham, MA, USA) for 5 min at 35 °C to dissociate the cells and cleave cell surface proteins, including ICAM-1. After adding 600 µL of trypsin-neutralizing solution (Lonza, Walkersville, MD, USA), the cells were collected by pipetting into Eppendorf tubes and spun down at 200× *g* for 5 min following washing with 1 mL of DPBS twice. The cell pellets were lysed by adding RLT buffer for RNA extraction. RNA samples were used to determine RV-A16 viral RNA copy numbers using qRT-PCR. 

### 2.8. Statistical Analysis

All results are shown as mean ± SD. One-way ANOVAs and t-tests were used for statistical analysis to determine the level of significant difference among all groups using Prism 9 software. Differences among groups were considered significant at a *p*-value less than 0.05.

## 3. Results

### 3.1. The Anionic Surfactant Lipid, PI, Inhibits RV-A16 Replication in Human AEC Cultures

First, we performed the time course experiments of lipid treatments with a viral challenge (pretreatment for 16 h before virus infection [RV16 + pre PI], simultaneous treatment and infection [RV16 + sim PI], and post-treatment 24 h after virus infection [RV16 + post PI] in differentiated ALI cultures of AEC from 3 donors (Appendix A). We found that only prophylaxis treatment for 16 h with PI induced the inhibitory effect (data are shown as mean ± SD % of RV16 infection alone and viral RNA copy number/well: RV16: 100.0% (24.6 ± 18.8 [×10^8^ viral copy numbers/well], RV16 + pre PI): 20.2 ± 10.6 (%) (3.9 ± 0.4 [×10^8^ viral copy numbers/well]), RV16 + sim PI: 110.8 ± 15.5 (%) (27.0 ± 17.5 [×10^8^ viral copy numbers/well]), and RV16 + post PI: 77.4 ± 21.4 (%) (17.7 ± 8.8 [×10^8^ viral copy numbers/well]). Next, we generated differentiated mucociliary ALI cultures from the basal AECs of four lung tissue donors. Replicate wells from each donor were subjected to the following treatments: (1) RV-A16 alone at 4 h and 48 h (shown as RV16 4 h and RV16, respectively), or (2) lipid pretreatment followed by RV-A16 infection (RV16 + POPG or PI or PC). Cultures were harvested at 4 h (input virus) and 48 h (virus progeny yield) after infection to evaluate viral replication, and the RV-A16 genomic RNA copy number was determined using qRT-PCR. We observed efficient viral replication (over 2-log increase in viral RNA) in the differentiated AECs (RV16 4 h: 0.09 ± 0.02 [×10^8^ viral copy numbers/well] vs. RV16 48 h: 12.4 ± 7.4 [×10^8^ viral copy numbers/well]) (Figure 1). Pretreatment with PI markedly reduced the viral RNA copy number at 48 h post-infection (p.i.) by 70% (RV16 + PI: 4.6 ± 3.2 [×10^8^ viral copy numbers/well], *p* < 0.05 in Figure 1A). In contrast, pretreatment with either POPG or PC did not inhibit viral replication (RV16 + POPG: 22.5 ± 11.2 [×10^8^ viral copy numbers/well], RV16 + PC: 18.5 ± 9.1 [×10^8^ viral copy numbers/well]) (Figure 1A). 

We next examined whether PI could inhibit the binding of RV-A16 to host cells. AECs from four different donors were incubated with RV-A16 at various doses (0.5–5 pfu ×10^5^/well) for 4 h at 4 °C with or without phospholipids added to the apical side of the cultures. After three washes with DPBS to remove the unbound virus, cell-associated RV-A16 RNA was quantified using qRT-PCR. The results have shown that RV-A16 can efficiently bind to AECs in a dose-dependent and saturable manner; however, the tested lipids did not interfere with virus binding to the cells (Appendix A). To further explore the possibility of the inhibition of virus entry into the host cells by PI, we used trypsin, a serine protease that cleaves peptides on the C-terminal side of lysine and arginine amino acid residues. When the virus internalization was inhibited by incubation at 4 °C, we observed about a one-log decrease in the viral RNA load after trypsin treatment compared to the no-trypsin control, thus indicating an efficient cleavage of the ICAM1 receptor with the bound virus from the cell surface (no-trypsin control cells (white square): 4.43 ± 0.8 [×10^7^ viral copy numbers/well] compared to trypsin-treated cells (white hexagon): 0.47 ± 0.11 [×10^7^ viral copy numbers/well] (*p* = 0.023) (Figure 1B). In contrast, after incubation at 35 °C, we found an overall increase in cell-associated virus in both the control and trypsinized cells, with a smaller decrease in the viral RNA load after trypsin treatment because most of the receptor-bound virus was internalized. We found no significant differences in cell-associated RV-A16 between PI-treated and untreated cells under any of the conditions tested. The results demonstrated that pre-treatment of cells with PI did not inhibit the virus internalization step (Figure 1B and Appendix A). 

To determine the possible cytotoxic effects of these lipids, we treated human AECs in ALI culture with PI or POPG and quantitated protein synthesis in cells and apical media using the ^3^H-leucine incorporation assay, as previously reported (Appendix A) [18,21]. We found that ^3^H-leucine incorporation was low and not significantly different between control cultures and cultures treated with any of the lipids.

We also tested whether POPG, PI, or PC had a direct virucidal effect using the modified method described in [21]. RV-A16 (10^5^ pfu) was incubated alone or with the lipids (PI [4 mg/mL], POPG [10 mg/mL], and PC [20 mg/mL]) in ALI growth medium at 35 °C for 4 h. The samples were serially diluted (10^−3^ to 10^−5^) to reduce the lipid concentrations and tested using plaque assay. Lipid treatments did not reduce RV-A16 plaque numbers (RV16: 38.3 ± 5.0, RV16 + POPG: 39.0 ± 4.6, RV16 + PI: 40.0 ± 2.0, and RV16 + PC: 37.7 ± 5.1), suggesting the lack of a virucidal effect of the lipids (Appendix A)

Lastly, we visualized RV-A16 infection in ALI cultures and examined whether PI reduced the number of virus-infected cells through immunofluorescent staining (Figure 1C). We found strong, virus-specific staining that was colocalized with dsRNA (virus replication intermediate) in a large number of cells infected with RV-A16 alone. However, PI treatment markedly reduced the number of virus-capsid and dsRNA-positive cells. These findings were in agreement with the results obtained through viral RNA quantification with RT-qPCR (Figure 1A).

### 3.2. Pulmonary Surfactant Lipids Reduce Antiviral and Proinflammatory Responses in Human AEC Cultures

We next examined whether the surfactant lipids affected the expression of important antiviral innate immunity genes in the context of RV-A16 infection of differentiated AECs. Specifically, we measured the expression of three epithelial antiviral genes, *MDA5*, *IRF-7,* and *IFN-lambda*, that are highly induced by virus infection but are expressed at a low level at baseline [5,37,38]. We found that pretreatment with PI significantly reduced RV-A16-stimulated *MDA5* expression (shown as the % of RV16 infection alone: RV16 + POPG: 76.4 ± 25.8 (%) (not significant (NS), RV16 + PI: 33.4 ± 18.1 (%) (*p* < 0.001)), and IRF-7 mRNA expression (% of RV16: RV16 + POPG: 89.0 ± 9.9 (%) (NS), RV16 + PI: 49.6 ± 14.4 (%) (*p* < 0.001), Figure 2A,B). Additionally, PI specifically inhibited *IFN-lambda* mRNA expression (% of RV16; RV16 + POPG: 122.9 ± 35.5 (%), RV16 + PI: 24.6 ± 15.2 (%) (*p* < 0.001), Figure 2C). PC did not modulate the expression of any of these genes.

We also examined the expression of *CXCL11*, a chemokine and airway epithelial cell biomarker of viral infections [3,39]. *CXCL11* was highly induced by RV-A16 in differentiated AECs and expressed at a low level at baseline. We found that treatment with PI markedly reduced *CXCL11* gene expression (% of RV16: RV16 + POPG: 83.3 ± 23.7 (%), RV16 + PI: 6.1 ± 4.1 (%) (*p* < 0.001), RV16 + PC: 104.6 ± 17.2 (%), Figure 3A). The inhibitory effect was also confirmed at the protein level, where we found that CXCL11 protein secretion was reduced by 37% and 80% by POPG and PI, respectively (% of RV16: RV16 + POPG: 63.1± 30.0 (%) (*p* < 0.05), RV16 + PI: 20.2 ± 6.0 (%) (*p* < 0.001), RV16 + PC: 135.4 ± 38.9 (%), Figure 3B).

Lastly, we measured the gene expression and protein secretion of IL-6, a cytokine that drives the inflammatory response to viral infections. The *IL6* gene and IL-6 protein were moderately expressed and secreted, respectively, at baseline and induced (about 2-fold) by RV-A16 infection. We found that both POPG and PI inhibited *IL6* gene expression (by 50% and 60%, respectively, Figure 4A) and protein production (by 57% and 80%, respectively, Figure 4B) after RV-A16 infection (shown as % of RV16, RV16 + POPG: 42.5 ± 8.5, RV16 + PI: 20.4 ± 5.9). PC did not affect IL-6 gene expression or protein secretion (Figure 4). 

We also compared the potency of POPG and PI on the inhibition of IL-6 production in RV-A16-infected human AECs. PI more efficiently reduced IL-6 protein production compared to POPG (IC_50_ of 0.12 mg/mL for PI vs. 2.93 mg/mL for POPG) (Figure 5).

### 3.3. Transcriptome-Wide Epithelial Responses to RV-A16 Infection Attenuated by PI Pretreatment

Given the significant downregulation of RV-A16 replication mediated by PI pretreatment and the inhibitory effects of PI and POPG on some antiviral and inflammatory responses, we next examined the effect of lipids on RV-A16-induced epithelial responses at a whole-transcriptome level. Specifically, we performed RNA-sequencing of the samples from the lipid treatment/RV-A16 infection experiments in AECs from the 3 different donors described above. Comparing control versus RV-A16 infected cultures, we found a dramatic epithelial response to infection with over 10,000 genes being up- and downregulated by infection (Figure 6A, Appendix A). The pathway enrichment analysis of RV-A16 differentially expressed genes (DEGs) produced expected pathway enrichments, including *IL-6 signaling, toll-like receptor signaling*, and *interferon signaling* (Appendix A).

To determine how lipid pretreatment influences responses to RV-A16 infection, we compared gene expression in RV-A16-infected cultures that were first pretreated with PC, POPG, and PI lipids (virus + lipid vs. virus) versus RV-A16 infection alone. We found that pretreating cells with PC (a lipid that did not inhibit viral replication nor antiviral/proinflammatory responses) before viral infection resulted in only a small number of DEGs compared to RV-A16 infection alone (up DEGs = 198, down DEGs = 194, Figure 6A). Moreover, the majority of these PC-mediated changes, while overlapping with RV-A16 DEGs (RV16 vs. Ctrl), enhanced the viral-mediated expression effect. In contrast, POPG pretreatment before RV-A16 infection resulted in >600 DEGs in comparison with RV-A16 infection alone, and 430 of these DEGs overlapped with RV-A16 DEGs (RV16 vs. Ctrl), 74% of which exhibited a direction of effect that *opposed* the direction of effect after RV-A16 infection (Figure 6A). Strikingly, we found 7388 DEGs in the “virus + PI vs. virus” infection analysis, with 5054 (68%) of these DEGs also being RV-A16 DEGs and having the direction of effect opposed to that of RV-A16 alone (Figure 6A and Appendix A). The PI treatment alone (PI vs. Ctrl) altered the gene expression of 5875 genes, 68% of which were also regulated by RV-A16 infection, but only 29% of them exhibited the opposite direction of effect. The strong opposing effects of PI treatment on the virus-induced responses are clearly seen when plotting log-fold changes (LFC) for RV-A16 DEGs between the “Virus vs. Ctrl” and “Virus + PI vs. Ctrl” comparisons (Figure 6B). 

To understand the effects of PI treatment at a pathway level, we analyzed the RV-A16 upregulated and downregulated genes, which were strongly attenuated by PI treatment (LFC > 0.5, *p* adj < 0.001, Appendix A). First, we analyzed the 1634 viral upregulated genes that were downregulated by PI and found 146 significant pathways, including those that are important to the anti-viral and proinflammatory response of the epithelium to viral infection: *the role of hyperchemokinemia in the pathogenesis of influenza, pathogen-induced cytokine storm signaling, and interferon signaling* (Figure 7A, Appendix A). The pathway schematic in Figure 7C shows genes underlying the interferon (IFN) signaling enrichment term, including fundamental IFN signaling genes: *SOCS1*, *STAT1*, *STAT2*, *IRF1*, *OAS1*, *IFITM1*, *IFITM2*, and *IFITM3*. In contrast, genes downregulated by RV-A16, which PI upregulated, were enriched for normal epithelial homeostatic metabolic pathways, including *xenobiotic metabolism signaling* and *ethanol degradation* (Figure 7B). 

We next explored the effects of RV-A16 infection and lipid treatment on epithelial cell remodeling, by testing whether the virus-affected genes opposed by PI were enriched for some cell-type markers (Figure 7D). The most prominent cell-type enrichment that we observed for genes upregulated by the virus and downregulated by PI (virUp:PIDown), was for mucus-secreting goblet cells. In contrast, we observed enrichments for club secretory cells, deutersomal cells, multiciliated cells, ionocytes, and neuroendocrine cells among genes downregulated by the virus but upregulated by PI pretreatment (virDown:PIUp). Taken together, these results suggest that viral induction of goblet cell metaplasia and the downregulation of some other common and rare epithelial cell types are opposed by PI lipid pretreatment.

### 3.4. PI Inhibits RV-A16 Mediated Regulation of Genes Critical for the Viral Replication Organelle (RO) Formation

Our gene expression studies suggest that some virus-induced host gene expression changes fundamental to viral replication may be modified by PI treatment. To investigate this further, we examined our RNA-seq data to determine whether PI treatment influenced the expression of 18 genes previously shown to play critical roles in the replication of picornaviruses via the formation and function of the replication organelles (ROs) [40,41,42,43] (Appendix A, Figure 8). Our analysis found that PI treatment in the absence of infection had only modest direct effects on the expression of the selected RO genes; however, it strongly inhibited virus-mediated effects on gene expression. Of the two PI4K genes, type3 phosphatidylinositol 4-kinase alpha (*PI4KA*) and beta (*PI4KB*), *PI4KB* is important for the formation of ROs and is required for RV replication within host cells [40,44]. PI markedly altered the ability of RV-A16 to downregulate *PI4KA* and induce *PI4KB* and *PI4K2B* gene expression (Figure 8A). PI also changed the expression of several *ACBD* and *ABCA* genes in the opposite direction compared to RV-A16 infection alone (Figure 8A). *LDLR* and *VLDLR* gene expression was also reduced to nearly the mock-infected level (Figure 8A). Collectively, PI treatment had a strong impact on the viral regulation of *PI4K* gene expression that can inhibit RO formation, and it reduced the expression of *LDLR* family genes that are crucial for intracellular cholesterol intake in host cells.

## 4. Discussion

In this report, we have shown that minor pulmonary surfactant phospholipid, PI, inhibits RV-A16 replication and virus-induced inflammatory responses in differentiated human AECs from multiple subjects. PI also reduced antiviral gene expression, including IFN-λ, which is particularly crucial in the innate immune response against RV infection by the host cells [45]. Treatment with POPG or PI alone did not inhibit *IRF-7* or *IFN-λ* gene expression. The majority of the innate immune response genes and many pro-inflammatory genes were expressed at low levels at baseline and were highly induced by virus infection; therefore, the observed inhibitory effects were mainly indirect and resulted from the inhibition of virus infection by other mechanisms. Both POPG and PI exist in the lung as natural compounds at very high concentrations and have potent protective effects against multiple respiratory viral infections (e.g., RSV, Influenza A viruses (H3N2-IAV, pH1N1-IAV), and SARS-CoV-2) [14,18,19,20,21] and viral elicited inflammation, as previously reported [14]. 

These lipids also antagonize inflammation by blocking multiple TLR activations in different human cells (alveolar macrophages, tracheal epithelial cells, and nasal epithelial cells) [14,17]. IL-6 is one of the major biomarkers induced by RV-A16 infection in the lungs and airways [46,47]. We determined that both PI and POPG lipids significantly inhibit IL-6 production in AEC apical medium (Figure 4) with the IC_50_ values (Figure 5) that were lower than their concentrations in vivo in alveoli (10 mg/mL for POPG and 5 mg/mL for PI). Interestingly, POPG failed to inhibit RV-A16 replication even though it attenuated virus-induced IL-6 and CXCL11 secretion at 48 h post-infection. Differentiated AECs secrete IL-6 at the baseline level, and it is likely that PI can inhibit IL-6 in both direct (via blocking TLR) and indirect (by inhibiting viral replication) ways. RV-A16 is a positive-strand RNA virus that produces dsRNA during replication in host cells [48,49]. The inhibitory effects of POPG on these cytokine productions very likely reflect the inhibitory efficacy of POPG on TLR3 activation by dsRNA.

As previously reported, POPG and PI antagonize multiple respiratory virus infections, including RSV and different strains of influenza A [14,18,19,20,21]. The major inhibitory mechanism of action of these lipids against these viruses is direct binding to RSV and IAV with high affinities, resulting in interfering with viral attachment to host cells [14,18,19,20,21]. We showed that, in contrast to RSV and IAV, PI does not inhibit RV-A16 attachment to the ICAM1 receptor (Appendix A) or virus entry into cultured AECs (Figure 1B). Both RSV and IAV are enveloped, negative-strand RNA viruses, while RV is a non-enveloped, positive-strand RNA virus that binds and enters the host cells via different mechanisms. In contrast to RSV and IAV, RV also modifies intracellular membranes to create replication organelles (RO) for efficient genome replication. We hypothesized that PI may inhibit the RO formation and function that are required for RV genome replication. These lipids do not have virucidal effects on RSV or IAV [14,18,19,20,21]. Similarly, PI also does not have a direct virucidal effect on RV-A16 (Appendix A). 

To understand the mechanisms of the inhibitory effect of PI against RV-A16 infection at the transcriptional level, we performed a whole-transcriptome gene expression analysis using RNA-seq after the infection of differentiated AEC. We found that PI treatment dramatically attenuated virus-induced gene expression (Figure 6), mainly due to the inability of the virus to initiate a productive infection and upregulate the antiviral and proinflammatory response genes. The pathway analyses of RV-A16 upregulated genes that were downregulated by PI identified multiple antiviral and proinflammatory response pathways, including “*pathogen-induced cytokine storm signaling” and “interferon signaling*”, whereas genes downregulated by RV-A16, which PI upregulated, were enriched for normal epithelial homeostatic metabolic pathways including, “*xenobiotic metabolism signaling*” and “*ethanol degradation*”. Accordingly, the cell-type enrichment analyses of virus-regulated genes opposed by PI treatment revealed that PI inhibited the viral induction of goblet cell metaplasia and the virus-induced downregulation of ciliated, club, and ionocyte cell types.

Notably, PI treatment altered the ability of RV-A16 to regulate the expression of some genes that play critical roles in the generation of ROs required for RV replication in host cells. The RO is a cellular compartment that is essential for the viral replication of many single-stranded RNA viruses, including rhinoviruses [40,41,42,43,44]. Specifically, viruses hijack human phosphatidylinositol 4-kinases (PI4K) to create specific phosphatidylinositol 4-phosphate (PIP4)-enriched organelles that are used for robust viral replication in host cells [40,41,42,43,44]. Products of several genes that are localized to the Golgi apparatus and ER [40,41,42,43] are crucial for the replication of RNA viruses [50,51,52]. PI4P is generated by PI4K that is categorized into type 2 (*PI4K2A* and *PI4K2B*) and type 3 (*PI4KA* and *PI4KB*, also known as *PI4KIIIα* and *PI4KIIIβ*, respectively) [44,53]. The hijacking of PI4Ks increases the phosphatidylinositol 4-phosphate (PI4P) level in RO, and particularly, PI4KB plays a critical role in the replication of picornaviruses, including rhinoviruses [44]. PI4KB controls the production of PI4P at the Golgi and trans-Golgi network (TGN) that regulates lipid transport by proteins, including oxysterol-binding protein (OSBP) [40,44]. ER membrane proteins, vesicle-associated membrane protein-associated protein isoforms A and B (VAP-A/B), are required for OSBP-mediated PI4P/cholesterol counter transport in ER [40,44,54]. The PI4P-cholesterol cycle is also regulated by OSBP and PI4K [55]. Protein acyl CoA-binding domain 3 (ACBD3) protein plays a key role in regulating PI4KB [56] and signaling and mediating PI4KB recruitment to ROs [44]. Free cholesterol in RO is required for enterovirus replication, and a lack of LDL in the culture medium or blocking cholesterol synthesis inhibits rhinovirus replication [57]. Cholesterol is a critical factor for rhinovirus replication and the viral life cycle [57].

We found that PI pretreatment followed by virus infection either attenuated or opposed the virus-induced transcriptional regulation of several key genes involved in the formation and function of ROs and cholesterol transport; however, PI treatment did not significantly affect the expression of those genes in the absence of infection. These findings suggest that, similarly to the observed inhibition of many other virus-induced genes and pathways, PI may affect the expression of those genes indirectly—possibly by inhibiting RV-A16 replication via transcription-independent mechanisms, such as (i) interactions with some virus proteins (e.g., 3A, 2B, 2C) [58] that are involved in cell membrane remodeling during RO formation; (ii) interactions with host RO proteins, such as ACBD3, SacI, and PI4KB; or (iii) interfering with PI4P/cholesterol transport at RO or (iv) as a result of virus replication inhibition by other, yet-unknown mechanism (Figure 8B). Additional studies are warranted to determine the mechanisms of PI interference with RO formation and function.

PI treatment had a strong impact on the viral regulation of *PI4K* gene expression that can inhibit RO formation, and reduced the expression of LDLR family genes that are crucial for intracellular cholesterol intake in host cells. PI4KA and PI4KB inhibitors are under development for clinical use [40,44]. PI4KB inhibitors are less toxic compared to PI4KA inhibitors, and they have the potential as antiviral agents for picornaviruses, including RVs, because PI4KB is required for their replication [40,44]. Some PI4K inhibitors have immunosuppressive or antiproliferative effects on lymphocytes in mice [44]. Recently, some new approaches to inhibit RV replication using DNAzyme and siRNA were shown to be effective in vitro [59,60]. These are novel approaches that may have great potential for use in clinical trials for the treatment of RV infections.

Our previous findings assure that PI does not have any toxic or immunosuppressive effects in an in vivo mouse model [14,15,19]. Treatment with PI provides an inhibitory effect on the virus-induced regulation of PI4K and PI4KB expression, and it can be administered intranasally and through inhalation therapy. Since PI is naturally present in the lungs, it is very unlikely that it can induce a host immune response. It has been safely applied to patients in clinical settings as a mixture of multiple surfactant components [14,15].

In summary, we report, in this study, that a minor pulmonary surfactant phospholipid PI has antiviral and anti-inflammatory effects against RV-A16 infection. PI might have the potential to be an efficient antiviral agent for the prophylaxis and treatment of RV infection and could be used as a compound tool to examine the mechanisms of viral replication.

## Figures and Tables

**Figure 1 viruses-15-00747-f001:**
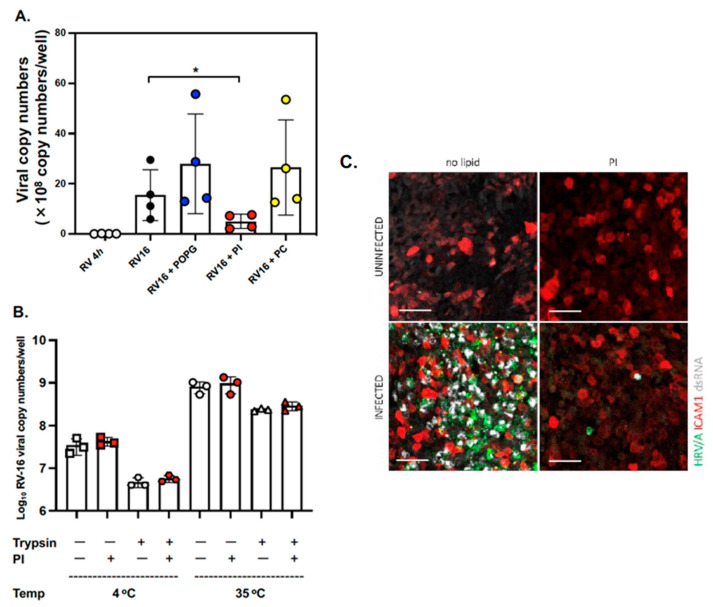
PI markedly inhibits RV-A16 replication but does not prevent viral internalization in AECs. (**A**) The data are from cells obtained from 4 different donors and are shown as mean ± SD from 3 independent experiments with duplicated wells in each experiment. * indicates, *p* < 0.05 in comparisons between RV16 alone and RV16 + PI. (**B**) The data are from cells obtained from 3 different donors and are shown as mean ± SD. The differentiated AECs were pre-treated (or not) for 16 h with PI (4 mg/mL) and infected apically with RV-A16 at 2.5 × 10^5^ pfu/well at 4 °C or 35 °C for 2 h. The cells were collected with or without trypsin/EDTA treatment before total RNA extraction to determine the viral RNA copy number using qRT-PCR. (**C**) Immunofluorescent staining of mock or RV-A16 infected AECs (48 h p.i.) that were treated (or not) with PI. Cells were stained with anti-ICAM1 antibodies (red) to confirm viral receptor expression, and anti-VP2 (green) and anti-dsRNA (white) antibodies to visualize virus-infected cells. The scale bar represents 50 μm.

**Figure 2 viruses-15-00747-f002:**
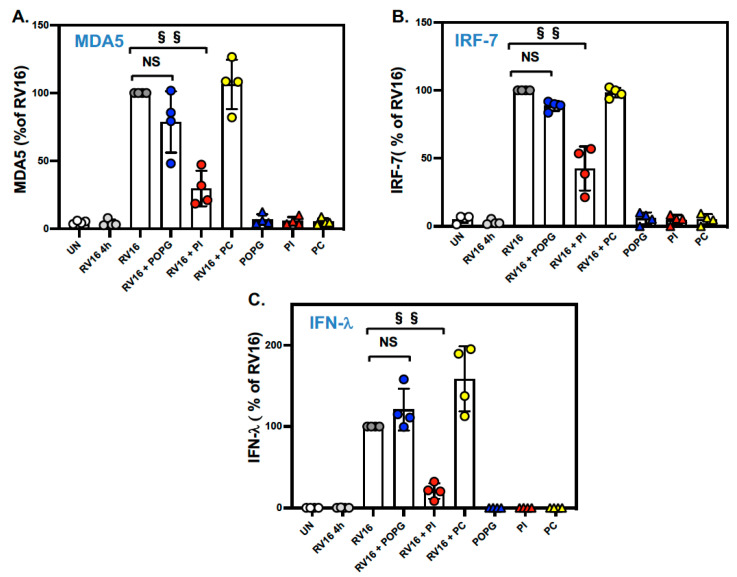
PI significantly reduced antiviral gene expression after the RV-A16 infection of AECs. (**A**) *MDA5* mRNA expression induced by RV-A16 infection. Treatment with PI reduced MDA5 mRNA expression by 65% (**RV16 + PI** as red circle) compared to RV-A16 alone (**RV16** as gray circle). (**B**) *IRF-7* mRNA expression. The treatment with PI (**RV16 + PI**) reduced IRF-7 expression by 50% in comparison to RV-A16 alone (**RV16**). (**C**) *IFN-lambda* expression at 48 h after RV-A16 challenge to AECs. The treatment with PI markedly attenuated *IFN-lambda* expression by 75% in comparison to RV16 alone. Both POPG (**RV16 + POPG** as blue circle) and PC (**RV16 + PC** as yellow circle) failed to impact *MDA5*, *IRF-7,* and *IFN-lambda* mRNA expression in panels (**B**,**C**). Lipid alone treatment were shown as **POPG** (blue triangle), **PI** (red triangle) and **PC** (yellow triangle). The data are from 4 different healthy control donors and are shown as mean ± SD from 3 independent experiments for each individual. §§ indicates, *p* < 0.001, and NS indicates not significant.

**Figure 3 viruses-15-00747-f003:**
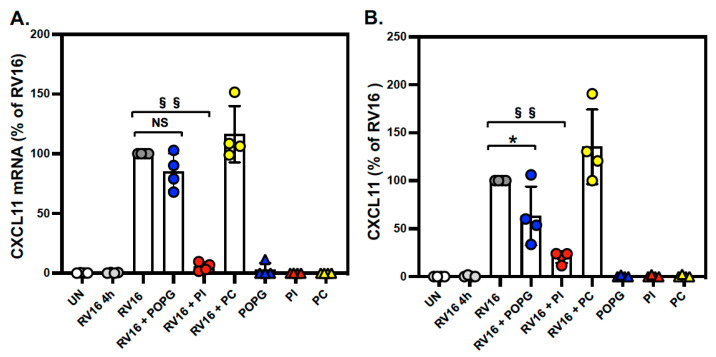
PI reduced *CXCL11* mRNA expression and protein secretion in RV-A16-infected AECs. Differentiated human AECs were treated with POPG, PI, or POPC, added to apical media, for 16 h prior to virus infection at an m.o.i. of 0.02 pfu/cell for 4 h at 35 °C, either with or without lipid treatments. Following DPBS wash to remove unbound virus at 4 h, the cells were cultured for 48 h, either with or without lipids. Groups consisted of uninfected (**UN:** white circle), viral binding and cell entry at 4 h (**RV16 4 h:** gray circle), the virus alone at 48 h p.i. (**RV16:** dark gray circle), the virus + POPG (10 mg/mL) (**RV16 + POPG:** bule circle), the virus + PI (4 mg/mL) (**RV16 + PI:** yellow circle), the virus + POPC (20 mg/mL) (**RV16 + PC**), and lipids alone (**POPG**, **PI**, and **PC**). (**A**) *CXCL11* mRNA expression and (**B**) CXCL-11 protein secretion induced by RV-A16 with or without lipid treatment. The lipid treatment alone were shown as **POPG** (blue triangle), **PI** (red triangle) and **PC** (yellow triangle). The data are from 4 different donors and are shown as mean ± SD from 3 independent experiments for each individual. * Indicates *p* < 0.05, §§ indicates *p* < 0.001, respectively. NS indicate not significant.

**Figure 4 viruses-15-00747-f004:**
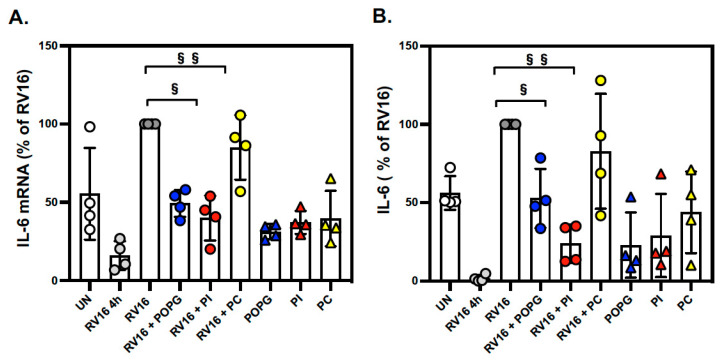
POPG and PI inhibit *IL6* mRNA expression and protein secretion in RV-A16-infected AECs. The cells were treated with lipids and infected as in Figure 3. (**A**) IL6 mRNA expression and (**B**) IL-6 protein secretion induced by RV-A16 infection with or without lipid treatment. § indicates *p* < 0.01 and §§ indicates *p* < 0.001, respectively. The data are shown as mean ± SD from 3 independent experiments for each subject (subject numbers n = 4). Groups consisted of uninfected (**UN:** white circle), viral binding and cell entry at 4 h (**RV16 4 h:** gray circle), the virus alone at 48 h p.i. (**RV16:** dark gray circle), the virus + POPG (**RV16 + POPG:** bule circle), the virus + PI (**RV16 + PI:** red circle), the virus + POPC (**RV16 + PC:** yellow circle), and lipids alone (**POPG:** blue triangle, **PI:** red triangle, and **PC:** yellow triangle).

**Figure 5 viruses-15-00747-f005:**
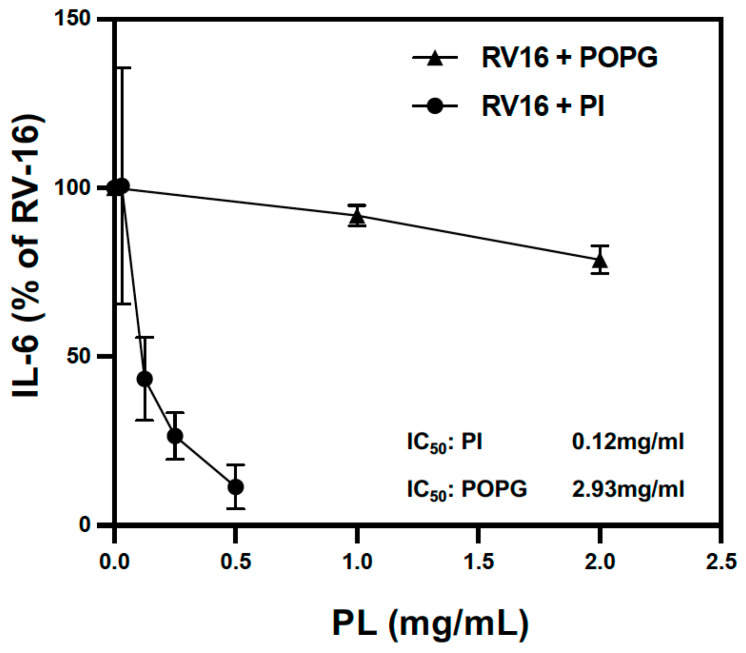
POPG and PI inhibited IL-6 production after RV-A16 infection in a dose-dependent manner. POPG and PI inhibited IL-6 secretion into apical media from AECs 48 h after RV-A16 infection in a dose-dependent manner. The half-maximal inhibitory concentration (IC_50_) of POPG is 2.93 mg/mL and 0.12 mg/mL for PI. The data are shown as mean ± SD from 3 independent experiments and duplicated wells in each experiment.

**Figure 6 viruses-15-00747-f006:**
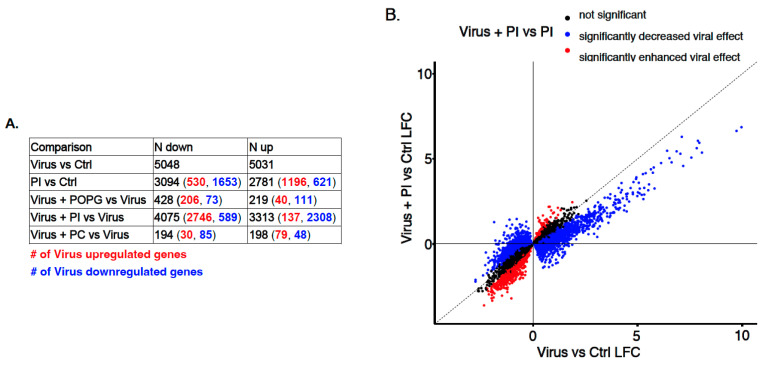
Differential gene expression analysis in RV-A16-infected human AECs with or without lipid treatment. (**A**) The number of up- and downregulated genes for the comparisons is indicated in the table. The numbers in parentheses show the number of upregulated (red) and downregulated (blue) RV-A16 response genes within the respective comparison. (**B**) Scatter plot of 10,079 DE RV-A16 response genes. RV-A16 response genes whose expression was opposed by PI are highlighted in blue (2746 + 2308 genes), and those enhanced by PI are highlighted in red (589 + 137 genes). Differential expression analyses were performed using the *lmerSeq* package.

**Figure 7 viruses-15-00747-f007:**
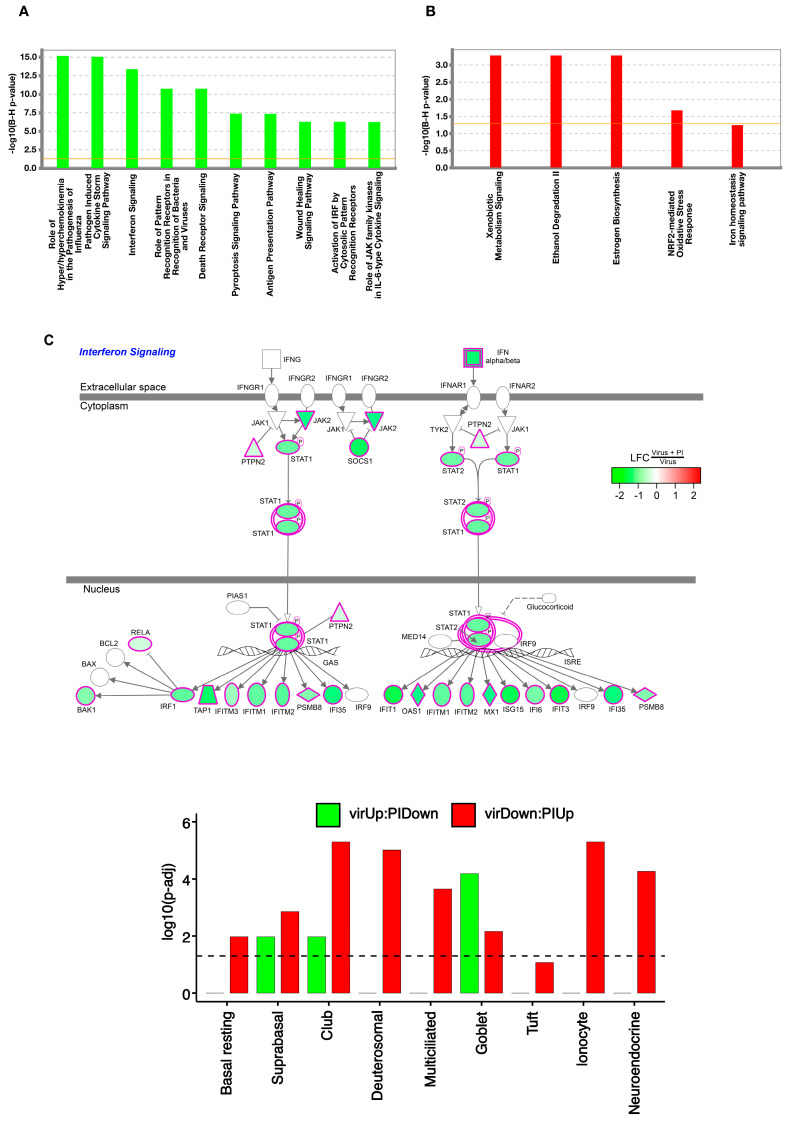
Pathway and cell–type enrichment analyses of RV–A16 response genes whose viral effect is opposed by PI. Barplots of enrichment scores of select IPA canonical pathways enriched for genes in the VirUp:PIDown (viral upregulated genes whose effect is opposed by PI) gene set (**A**) and VirDown:PIUp (viral downregulated genes whose effect is opposed by PI) gene set (**B**). Enrichment scores are represented as -log10 of Benjamini–Hochberg (BH) adjusted *p*-values obtained from IPA. (**C**). Cell diagram of the IPA interferon signaling pathway, with node colors representing the logFC in gene expression between RV-A16 + PI and RV-A16 cultures. (**D**). Barplot of enrichment scores for airway epithelial cell-type gene signatures within VirUp:PIDown (green) and VirDown:PiUp (red) gene sets. Enrichment scores were represented as –log10 of BH–adjusted *p*-values obtained from the hypergeometric test.

**Figure 8 viruses-15-00747-f008:**
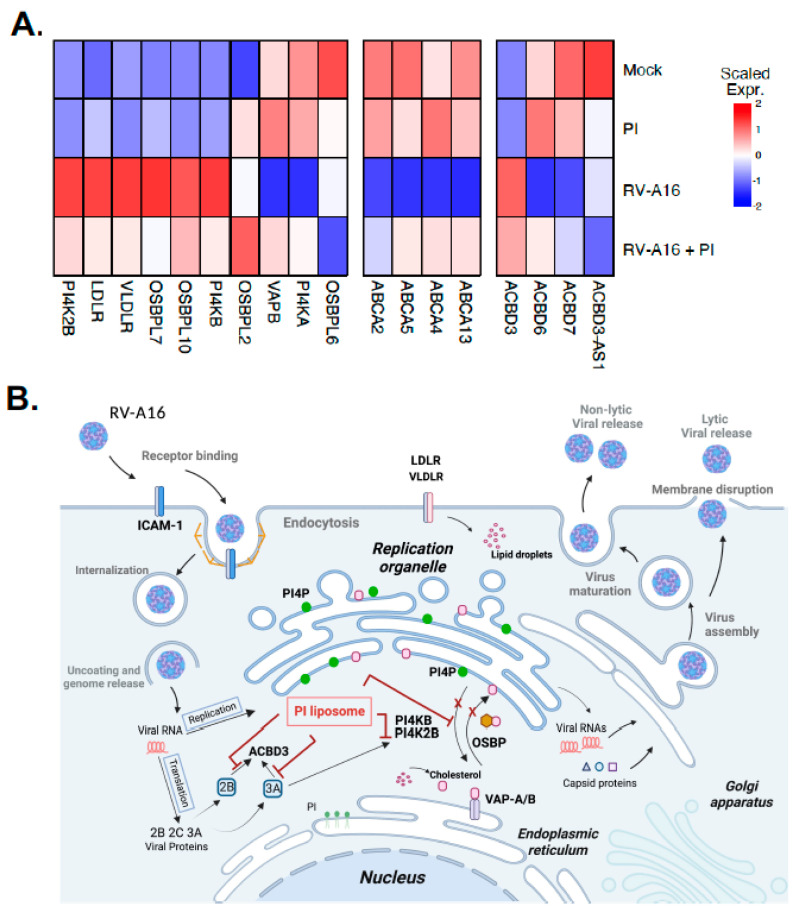
PI attenuated the RV–A16-mediated regulation of genes involved in replication organelle (RO) formation and function. (**A**) Heat map of scaled average normalized expression for each condition: (1) uninfected (**mock**), (2) treated with PI alone (**PI**), (3) virus-infected (**RV–A16**), and (4) virus-infected and treated with PI (**RV16 + PI**) for RV–A16–regulated genes involved in RO formation and function. (**B**) Schematic of the RV-A16 replication cycle and the possible mechanisms of action of the PI liposome on RO formation and function.

## Data Availability

All data relevant to the study are included in the article and supplementary Materials.

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
