# Peer review of "Anionic Pulmonary Surfactant Lipid Treatment Inhibits Rhinovirus A Infection of the Human Airway Epithelium"

_viruses, 2023, doi:10.3390/v15030747_

Round 1

Reviewer 1 Report

In this manuscript, Numata et al., expand upon prior work indicating antiviral and immunosuppressive effects of pulmonary surfactant lipids to investigate the impact of POPG, PC, and PI during rhinovirus infection in a differentiated model of human airway epithelium.

While the data are potentially interesting, the impact is significantly limited by (1) the use of one rhinovirus (RV-A16), (2) the limited focus only on one time point after infection (48hrs), and (3) investigation of pre-, but not post-, treatment with PI. Further, and perhaps most importantly, (4) the authors do not consider the differences in infection (as suggested in Fig 1) in their subsequent cytokine analyses. Thus, it is impossible to understand whether PI restricts viral uptake / replication through some undefined mechanism which then results in a muted host response and reduced airway remodeling - or - if PI does (as the authors seem to suggest, especially at lines 356, 382 etc.) have a direct impact on proinflammatory / interferon responses. Consideration of the numbers of infected cells, for example, during data interpretation or revising their discussion of these data is warranted. Finally, (5) there are no follow up studies of the RNAseq data at the protein level. Inclusion of western blot analysis, flow cytometry, or imaging of key host factors here would strengthen the conclusions.

Additional minor comments are listed below.

- Since MOI is not accurately determined in these cultures, please report the total PFU applied and culture surface area.

-The methods are a bit confusing as the phospholipids were added “in apical media”, but yet these cultures are at air-liquid interface. What volume that was applied to the apical surface?  Was the same volume (vehicle) applied to conditions that did not get phospholipids?

-In section 3.1 (line 168) significance is noted as p < 0.05, however in the figure legend it says p<0.001.  Please clarify.

-I appreciate that the authors report the actual titers in the text; however, for example, in Fig 1 (and throughout), all the values in the RV16 condition are set to 100%.  This doesn’t seem to capture the variation noted in the text (12.4+/-7.4 x 10^8).  Please elaborate on how these data were transformed.

-The authors note that 3H-leucine incorporation was low – does this indicate that the dynamic range in this assay to look at potential cytotoxic effects was minimal and potentially unable to resolve a negative effect on cell viability?

Reviewer 2 Report

his study from Numata et al examine the impact of dioleoyl-phosphotidylinositol (PI) or palmitoyl-oleoyl-phosphatidylglycerol (POPG) on replication of Human rhinovirus 16 (HRV16) I in human airway epithelial cells (AEC) grown in air liquid interface (ALI) cultures. Infection of cells pre-treated with PI for 16 hours reduced viral RNA copy number significantly while POPG did not have an effect that reached statistical significance. This wasn’t due to reduced cell metabolism, direct inactivation of the virus or a block in virus binding to the cell receptor. Gene expression analysis revealed PI reduced mRNA expression levels of MDA5, IRF-7 and IFN-lambda, CXCL11 and IL-6 in infected cells, with subsequent analysis extending this to protein levels for CXCL11 and IL-6. POPG reduced protein and mRNA levels of IL-6 but only protein levels for CXCL11. RNAseq revealed that pre-treatment with PI significantly attenuated the typical transcriptional response seen upon HRV16 infection of AECs. Gene ontology analysis of the transcripts alternatively regulated by HRV16 in the presence of absence of PI treatment reveals general inhibition of the antiviral response and along with genes associated with goblet cell differentiation. Additional analysis showed that PI prevented virus-induced modification of expression levels of a variety of genes involved in replication.

The results show a clear inhibitory effect of PI on HRV16 replication in AECs. This is associated with significant changes in gene expression.  Somewhat surprisingly this is not associated with a heightened antiviral response and instead the authors propose that the reduction is more likely due to PI blocking viral up/downregulation of genes involved in forming replication organelles, although experiments are not done to confirm this. One important question left unanswered is if PI has a general inhibitory effect on the antiviral response- for example in response to other viruses or poly I:C treatment. Another surprising result from this study was the inhibition of HRV replication by PI appears to utilize a different mechanism from what was described for RSV and Influenza virus. Do the authors think that the effect identified here may also contribute to inhibition of these enveloped viruses? A more thorough comparison of the results presented here with the results from those earlier studies is warranted. Another question that could be pursued is how long a pretreatment is necessary for inhibition and would administration at the time of infection give the same result? Have the authors looked at treatments shorter than 16 hours or varied the concentration of PI?

Reviewer 3 Report

Summary 

The authors outline the complexity of preventing rhinovirus (RV) infection due to the multiplicity of species and serotypes, as well as effective therapies in order to prevent infection complications, like asthma exacerbations. They reference pulmonary surfactant, which helps to regulate lung innate immunity. They reference POPG and PI as inflammation regulators (TLR antagonists) with antiviral activity against RSV and influenza A. They examined their effects against RV A16 in primary human airway epithelial cells (AEC) using an air-liquid interface. They observed that in RV-A16 infected AEC, PI reduced viral RNA copy number by 80% and down regulated specific antiviral genes by 55-65%. POPG showed only modest similar effects. Both POPG and PI inhibited IL6 and CXCL11 gene expression and protein secretion. PI substantially attenuated gene expression induced by RV-A16. It further inhibited viral induction of goblet cells meta plasma and viral-mediated down regulation of ciliated and other cell types, and altered viral ability to regulate PI4K and other genes critical to formation and functioning of replication organelles needed for RV replication in host cells. They conclude that PI may be useful as a potent, non-toxic antiviral agent for RV infection prophylaxis and therapy. 

General comments 

The manuscript background makes a case as to why RV infection is important, specifically its contribution to asthma exacerbation in children, as well as the foundational work relevant to their current investigation, all of which is appropriately referenced. The methodology section is sufficiently detailed in stepwise fashion that their work should be replicable by other laboratory scientists. The statistical analysis plan, though brief, appears appropriate to both the outcomes and study design. The gene and chemokine inhibitory effect of PI are well documented and consistent. The experiments showing the attenuation of transcriptome-wide epithelial responses by PI pretreatment are particularly elegant. 

The implications for both prophylactic and therapeutic effects of PI and POPG, including the limitations of the latter, are clearly detailed using these in vitro systems, including the inhibitory effects of PI on genes essential to RV replication. That PI pretreatment did not affect genes involved in RO expression in the absence of infection may indicate a possible safety signal, to be followed up in future in vivo studies. As such, the conclusions are well justified, and further study is warranted.  

Specific comments 

None. 

Reviewer 4 Report

The manuscript demonstrates that one of the component of pulmonary surfactant phopholipds, phosphatidylinositol (PI) inhibits rhinovirus replication and rhinovirus-induced antiviral and inflammatory cytokines.  It was found that PI-induced changes is not due to inhibition of virus binding to the cells.  The authors further show that PI also does not have direct antiviral effect.  The authors go on to conduct RNAseq analysis to explore the signaling pathways that is altered by PI in RV-infected and uninfected cells.  RNAseq analysis demonstrates that almost all the RV-induced signaling is inhibited by PI.  Most importantly, PIP4kinase pathway that plays role in virus replication was found to be attenuated in PI treated RV-infected cells.  All the observed inhibitory effect of PI could be due simply to inhibition of virus internalization or virus entry into the cells.  PIP kinases are involved not only at later step of the virus replication but also for entry of virus. The paper is well-written but lacks the in depth experimental evidence on how PI affects rhinovirus replication.

Major comments

1.      The method used to determine the binding determines both bound and internalized virus.   The authors must use flow cytometry to detect bound versus internalized virus

2.      In fact, flow cytometry can also indicate what kind of cells are preferred by virus to gain entry into the epithelium

3.      Although the virus binds to the cell, if the virus does not enter the cell, almost all the signaling induced by virus will be inhibited and this is what observed by RNAseq anlayses

4.      If the observed effects are not due to attenuated entry of the virus into the cells, authors should consider providing experimental evidence on the role of PIP4 kinase in PI-inhibited RV replication

5.      Rhinovirus has been shown to signal through TLR2 and affect cellular responses.  Since authors have previously shown that PI inhibits TLR pathways, authors should examine whether the observed effects are due to inhibition of TLR2 signaling  

Reviewer 5 Report

With interest, I read the manuscript viruses-2188315, an interesting experimental work by experienced researchers. Now, when coronavirus dominated everythIng, we should not forget about important pathogens always present with us. Thus, I am grateful to the Authors for this paper.

I have only minor/facultative comments:

1.     Statistical analysis. What do you mean “single” ANOVA. Besides, what was post-host test following ANOVA?

2.     The names of the genes must be written in italics, also in the graphs.

3.     Figures 6 and 7 contain some details difficult to be seen or read. Please, correct.

4.     Some other experimental anti-RV therapies should be mentioned in the Discussion, e.g. antisense strategies (PMID: 30114391, 14670593).

Round 2

Reviewer 1 Report

The authors have improved the manuscript through inclusion of new data (Fig 1B) and also through added discussion on the impact of PI on infection. I have just a few minor comments below:

1) The language in the first sentence of the results is vauge- especially that 16hr pre-treatment was "optimal".  While this result is expected, it warrants more specific details even if the data is not shown.  Was ANY effect observed when PI is added simultaneously or after infection? In the author response the statement is made: "We found significant inhibitory effect of PI against RV-A16 only after pre-treatment for 16hrs."  This implies that a statistical evaluation was done and if so, such a statement would help clarify this section.  In addition, it is not clear in this first line what cell model is being used.  

2) The term "under expressed" is used in several places in the manuscript but this does not sound correct.  Do you mean that expression was reduced compared to something else??

3) In the discussion, I would delete "that have attachment proteins" from line 465 and modify this sentence to read: "Both RSV and IAV are enveloped, negative-strand RNA viruses while RV is a non-enveloped, positive strand RNA virus that binds..."

Reviewer 2 Report

The revised manuscript have responded to the points I raised. They have added information stating that pretreatment with PI provides the strongest proviral effect. However, this data is not shown and, I think, should be presented to the reader for evaluation. They had expanded their discussion of the different mechanisms by which PI inhibits enveloped vs non-enveloped viruses such as HRV. They have also addressed my questions regarding the general effect of PI on the antiviral response.

Reviewer 4 Report

Major comments were not addressed by additional experiments.  It is extremely important to determine the bound and internalized RV by using appropriate method.  Determining binding at 4 C is not appropriate method.  Moreover the immunofluorescence detection of dsRNA does not provide anymore info on the bound versus internalized RV.  If the RV is not internalized then there will not be any intracellular dsRNA.  

1.  In order to determine bound vs internalized RV one can also use immunofluorescence.  First immunostain the surface bound live virus, then fix and permeabilize the cells to detect internalized virus. 

2. Figure 1B does not mention the what colour represents what antibody used. 
